# Novel Graphene/In_2_O_3_ Nanocubes Preparation and Selective Electrochemical Detection for L-Lysine of *Camellia nitidissima* Chi

**DOI:** 10.3390/ma13081999

**Published:** 2020-04-24

**Authors:** Jinsheng Cheng, Sheng Zhong, Weihong Wan, Xiaoyuan Chen, Ali Chen, Ying Cheng

**Affiliations:** 1Henry-Fork School of Food Sciences, Shaoguan University, Shaoguan 512005, China; weihongsgu@163.com (W.W.); xychensgu@126.com (X.C.); 2Shipai Branch, Dongguan Environmental Protection Bureau, Dongguan 523330, China; jasonwow@163.com; 3Foshan Qionglu Health Tech. Ltd., Foshan 528000, China; yingchenggd@163.com; 4School of Pharmacy, Guangdong Pharmaceutical University, Guangzhou 510006, China; chenali2004@163.com

**Keywords:** graphene/In_2_O_3_ cubes, L-Lysine, detection, *Camellia nitidissima* Chi

## Abstract

In this work, novel graphene/In_2_O_3_ (GR/In_2_O_3_) nanocubes were prepared via one-pot solvothermal treatment, reduction reaction, and successive annealing technology at 600 °C step by step. Interestingly, In_2_O_3_ with featured cubic morphology was observed to grow on multi-layered graphene nanosheets, forming novel GR/In_2_O_3_ nanocubes. The resulting nanocomposites were characterized using transmission electron microscopy (TEM), scanning electron microscopy (SEM), X-ray diffraction spectroscopy (XRD), etc. Further investigations demonstrated that a selective electrochemical sensor based on the prepared GR/In_2_O_3_ nanocubes can be achieved. By using the prepared GR/In_2_O_3_-based electrochemical sensor, the enantioselective and chem-selective performance, as well as the optimal conditions for L-Lysine detection in *Camellia nitidissima* Chi, were evaluated. The experimental results revealed that the GR/In_2_O_3_ nanocube-based electrochemical sensor showed good chiral recognition features for L-lysine in *Camellia nitidissima* Chi with a linear range of 0.23–30 μmol·L^−1^, together with selectivity and anti-interference properties for other different amino acids in *Camellia nitidissima* Chi.

## 1. Introduction

In the past years, graphene has gained various interests in different fields [1,2,3,4,5]. This novel 2D carbon-based nanomaterial possesses superior chemical, physical, and electrical properties [6,7,8,9,10]. A variety of intriguing graphene-based nanocomposites with different morphologies, for instance, nanoparticles [11], nanoshuttles [3], nanorods [12], nanofibers [13], nanosheets [14], metal organic frameworks (MOFs) [15], and quantum dots [16], etc., were prepared with different applications. Recently, some researchers studied various functional graphene nanocomposite-based electrochemical sensors for different applications. For example, our team developed a sodium dodecyl benzene sulphonate (SDBS) functionalized graphene nanosheet-based electrochemical biosensor, which showed excellent electrocatalytic performance toward the reduction of H_2_O_2_ with fast response, wide linear range, high sensitivity, and good stability [17]. Kang et al. employed thermally split graphene oxide, both of which exhibited similar excellent direct electrochemistry of glucose oxidase (GOD) [18]. However, there were few reports concerning the synthesis and application of transition-metal nanocubes/graphene nanocomposites.

*Camellia nitidissima* Chi, a widely used plant in Southern China, is known as the “Giant Panda of Botany” [3]. In 2010, the Ministry of Health of the People’s Republic of China approved *Camellia nitidissima* Chi as a new resource of food (2010 Chinese Ministry of Health Announcement No. 9). Investigations revealed that *Camellia nitidissima* Chi contains rich amino acids, vitamin C, gross sugar and protein, etc. Moreover, the contents of some typical essential amino acids (EAAs) in cultivated *Camellia nitidissima* Chi, for example, Leu, Thr, Val, and Ile, were higher than those in the Food and Agriculture Organization of the United Nations (FAO)/World Health Organization (WHO) reference contents [19]. Our previous research also showed that by using graphene nanoshuttle matrix-based matrix-assisted laser desorption/ionization time of flight mass spectrometry (MALDI-TOF MS), fifteen kinds of amino acid ingredients, including L-lysine (L-Lys), L-aspartic acid (L-Asp), L-threonine (L-Thr), L-serine (L-Ser), L-glutamic acid (L-Gly), etc., can be detected readily in *Camellia nitidissima* Chi with strong peaks and good discrimination [3]. Similar results were also observed in our other work. By combining with gas chromatography-mass spectrometry (GC-MS), the combined stir bar sorptive extraction (SBSE)/GC-MS technology with linear graphene nanocomposite coating can detect seventeen kinds of amino acids in *Camellia nitidissima* Chi seeds, including Ala, Gly, Thr, Ser, Val, Leu, Ile, Cys, Pro, Met, Asp, Phe, Glu, Lys, Tyr, His, and Arg [13].

The research on the homochirality of basic biological molecules such as amino acids is still an intriguing puzzle in modern sciences [20]. Research on chiral recognition and chiral selection of some kinds of amino acids [21] could provide fundamental scientific thought towards resolving the puzzle regarding some kinds of homochirality and other important applications. However, until now, few reports have focused on the chiral recognition of a single kind of amino acid, for example, L-Lys, in *Camellia nitidissima* Chi.

In this study, using one-pot solvothermal treatment, together with reduction and annealing treatments in turn, we successfully prepared novel graphene/In_2_O_3_ (GR/In_2_O_3_) nanocubes. The prepared nanocomposites were characterized by transmission electron microscopy, scanning electron microscopy, X-ray diffraction spectroscopy, etc. A selective electrochemical sensor for L-Lys in *Camellia nitidissima* Chi based on GR/In_2_O_3_ nanocubes was designed and applied. In a linear range of 0.23–30 μmol·L^−1^, the GR/In_2_O_3_-based electrochemical sensors illustrated good chiral recognition of L-Lys in *Camellia nitidissima* Chi with a strong selective signal.

## 2. Results and Discussion

### 2.1. Preparation and Characterization of GR/In_2_O_3_ Nanocubes

Until now, there have been few reports concerning the preparation and application of GR/In_2_O_3_ nanocubes, which would combine both superior electrical properties of graphene and In_2_O_3_ nanocubes. In this work, we prepared novel GR/In_2_O_3_ nanocubes readily by solvothermal, reduction, and annealing procedures step by step. After vacuum drying, the GR/In_2_O_3_ nanocubes were finally afforded (Scheme 1).

The resulting nanocomposites were characterized by TEM, scanning electron microscope (SEM), and XRD, etc. The TEM images (Figure 1a) display a view of the prepared nanocomposites, and the cubic structured nanocomposites are evenly distributed on graphene nanosheets (Figure 1b). The HRTEM image shown in Figure 1c indicates clear evidence of polycrystalline structures of graphene nanosheet-wrapped In_2_O_3_, which shows the lattice spacing of 0.28 nm, similar to the In_2_O_3_ (222) lattice spacing [22]. The inserted image in Figure 1c (upper right corner) also shows the electron diffraction pattern of the graphene nanosheet-coated In_2_O_3_ nanocomposites. Another HRTEM image clearly demonstrates that the featured wrinkled multi-layered graphene supports uniformly distributed cubic structures (Figure 1d). The cubic structures on the graphene nanosheets had a diameter of about 30–70 nm.

The SEM image confirms the cubic structures of In_2_O_3_ on graphene nanosheets (curve a in Appendix A). The diameter of the synthesized In_2_O_3_ nanocubes had a size range of about 30–70 nm (see particle size distribution image, curve b in Appendix A), which was in accordance with the results observed in Figure 1.

Further evidence for the synthesized GR/In_2_O_3_ nanocubes was supplied by XRD. From the XRD patterns in Figure 2A (curve c), it can be observed that graphene nanosheets give a weak characteristic peak at 2θ = 24.7°, which conforms with the literature’s results of graphene peaks (002) [13]. The peaks at 2θ = 22.26°, 31.68°, 39.08°, 45.42°, 51.56°, and 56.48° can be assigned to (200), (220), (222), (400), (420), and (422) crystalline plane diffraction peaks of In(OH)_3_ (JCPDS no. 85–1338, curve b in Figure 2A). Meanwhile, a strong peak of In_2_O_3_ (222) was observed at 30.58°, suggesting the characteristics of In_2_O_3_. The other typical peaks at 21.5°, 35.47°, 45.69°, 51.03°, and 60.68° were attributed to (211), (400), (431), (440), and (622) crystalline plane diffraction peaks of In_2_O_3_, respectively (JCPDS no. 65–3170, curve c in Figure 2A). An energy dispersive X-ray spectroscopy (EDS) taken from a random assembly of GR/In_2_O_3_ nanocubes also identified the In_2_O_3_ nature supported on graphene (Figure 2B).

### 2.2. The Mechanism of the Prepared GR/In_2_O_3_ Nanocubes

In this work, we used graphene oxide (GO,) indium acetylacetonate, and NaOH as the starting materials, and the mixed solvent, EtOH-H_2_O-cyclohexane-oleic acid, was used. After a solvothermal procedure at 200 °C for 20 h, the intermediate GO/In(III) was given. In these procedures, SN_2_ and hydrolysis reactions occurred in turn, resulting in a GO-supported In(OH)_3_ nucleus (GO/In(OH)_3_). After the reduction reaction at room temperature in the presence of Na powder, the nucleus of GR/In(OH)_3_ (reduction type of GO/In(OH)_3_ nucleus) was given. From the inserted TEM image on the upper side of GR/In(OH)_3_ intermediate, we could observe many dense accumulations, and each dense accumulation showed a lot of GR/In_2_O_3_ nuclei gathering together, giving the embryonic form of GR/In_2_O_3_ cubes. In the final step, the prepared product was annealed at 600 °C for 2 h under Ar flow. In this procedure, a self-assembly of the original generated GR/In_2_O_3_ nucleus occurred readily, resulting in the final product: GR/In_2_O_3_ nanocubes (Figure 3).

### 2.3. Response of Amino Acids of Camellia nitidissima Chi Using GR/In_2_O_3_ Nanocube-Based Electrochemical Sensors

A selective electrochemical sensor for L-Lys in *Camellia nitidissima* Chi based on GR/In_2_O_3_ nanocubes was prepared readily. Coated with *Rheinheimera speciales*, a typical L-lysine-ε-oxidase, the designed electrochemical sensor was expected to have a good response to L-Lys. As described in Scheme 2, by modifying with *Rheinheimera speciales* and horseradish peroxidase etc., a mixed solution of GR/In_2_O_3_-RHS-HRP-AQ-55D was given, which was dropped on the polished glass carbon electrode (GCE) surface, forming a GR/In_2_O_3_-RH-GCE electrode. By using a three-electrode system, the concentrations of the amino acids in *Camellia nitidissima* Chi could be determined by the recorded current change.

The prepared GR/In_2_O_3_ nanocube-based electrochemical sensor was used to detect the selectivity of L-Lys (Scheme 2). A phosphate buffer solution with a pH of 7.8 was used as the supporting electrolyte background solution. The operating voltage was +0.85 V, and after 200 s, the current was stabilized and the concentrations of the amino acids in *Camellia nitidissima* Chi could be determined readily. Our previous research showed that there are fifteen kinds of amino acid ingredients, including L-Lys, L-Asp, L-Thr, L-Ser, L-Glu, L-Pro, and L-Gly, etc., which can be detected readily in the leaves of *Camellia nitidissima* Chi with strong peaks and good discrimination [3]. Among these, the analytical result of L-Lys was 0.45%, which was in agreement with the literature report [23]. For comparison, D-Lys was added to the solution with the same concentration (0.45%) to test the chiral recognition properties of the prepared GR/In_2_O_3_ nanocube-based electrochemical sensor.

Moreover, a series of experiments was conducted to test the linear range of the prepared electrochemical sensors for L-Lys. As shown in Appendix A, within a good concentration range, the value of the response current of L-Lys increased with the increasement of the concentration of L-Lys. When the concentration increased to 30 µmol·L^−1^, the response current would stop rising and start falling. Within the concentration period of 30–70 μmol·L^−1^, the response current decreased gradually with the concentration growth of L-lysine. The possible reasons might be that the active sites of the prepared GR/In_2_O_3_ nanocubes were gradually filled by L-Lys, and the adsorption tends to be saturated above the concentration of 30 μmol·L^−1^ of L-Lys. Therefore, the response current tends to be stable within a concentration range of 30–70 μmol·L^−1^. Investigations illustrated that there was a good linear relationship between the concentration of L-Lys and the response current within the range of 0.23–30 μmol·L^−1^.

The chiral recognition, selectivity, and anti-interference properties for L-Lys of the prepared electrochemical sensors based on GR/In_2_O_3_ nanocubes were observed by using the amperometric method. After the response current was stabilized with a background solution, free amino acid extraction of *Camellia nitidissima* Chi leaves (with an extra addition of D-lysine (D-Lys) at a concentration of 0.45% to test the chiral recognition of the prepared sensor) was then tested. As illustrated in Figure 4, the current response signal of the i-t graph had an obvious step-like change. Previous research indicated that there were 15 kinds of free amino acids in *Camellia nitidissima* Chi leaves [3], while in this study, as demonstrated in Figure 4, the strongest current signal was L-Lys. For other kinds of amino acids in *Camellia nitidissima* Chi leaves, e.g., L-Asp, L-Thr, L-Ser, L-Glu, L-Pro and L-Gly, L-Ala, L-Val, L-Ile, L-Leu, L-Tyr, L-Phe, and L-His, etc., no obvious signals were detected by the prepared sensor, indicating that the prepared electrochemical sensor based on GR/In_2_O_3_ nanocubes had good selectivity and anti-interference properties. Meanwhile, we could observe a feeble signal at about 500 V, which can be assigned to the response signal of L-Arg, which might due to L-Arg having a similar structure to that of L-Lys. In addition, the oxidation peak positions of both L-Arg and L-Lys were very close, so it is difficult to separate two such kinds of amino acids. However, the current change caused by L-Arg was much smaller than that of L-Lys, indicating that the prepared sensor had good selectivity and anti-interference properties.

As a comparison, to test the chiral recognition of the prepared sensor, D-Lys with the same detected concentration of L-Lys in the sample of *Camellia nitidissima* Chi was added. From Figure 4, we can observe that D-Lys had a weak electrochemical signal. Moreover, the signal intensity was much weaker than that of L-Lys, revealing that the prepared sensor had a good chiral recognition property.

Meanwhile, we also studied size non-uniformity affects of the prepared sensor. As shown in Appendix A, when GR/In_2_O_3_ nanocubes with non-uniformity size range of 20–100 nm was obtained (curve a), besides peaks of L-Lys, D-Lys and L-Arg interferential peaks were also observed.

Furthermore, differential pulse voltammetry (DPV) responses of L-Lys by GR/In_2_O_3_ nanocube-based electrochemical sensor were also detected. As shown in Figure 5, after using the GR/In_2_O_3_ nanocube-based electrochemical sensor, the current response values of the L-Lys on the electrodes were significantly negatively shifted (as shown in Figure 5a, when an electrode without GR/In_2_O_3_ nanocubes was used, the value was 0.84 V, while as illustrated in Figure 5b, the oxidation peak potential of the modified electrode changed to 0.72 V, together with much improved peak intensity), which indicates that after being modified with GR/In_2_O_3_ nanocubes, the adsorption amount and response speed of L-Lyse were significantly improved.

## 3. Experimental Details

### 3.1. Materials and Characterization

*Camellia nitidissima* Chi leaves were collected from Fangcheng Port City, Guangxi Zhuang Autonomous Region, China. Graphene oxide (GO) was prepared using the modified Hummers method [24,25]. Graphite powder (99.99995%, 325 mesh) and indium acetylacetonate (99%) were purchased from Alfa Aesar (Tianjin), Tianjin, China. L-lysine (≥98%) and D-lysine (≥98%) were purchased from Sigma Aldrich, Shanghai, China, and *Rheinheimera speciales*, bovine serum protein, and horseradish peroxidase were purchased from Shenggong (Shanghai, China) Co. Ltd. All solvents and other reagents were purchased from Beijing Chemicals Co. Ltd (Beijing, China) as analytical-grade products.

The powder X-ray diffraction (XRD) measurements of the samples were recorded on a Bruker D8-Advance X-ray powder diffractometer (Bruker Scientific Instruments Hong Kong Co. Ltd., Hong Kong, China) using Cu Kα radiation (λ = 1.5406 Å) with scattering angles (2θ) of 8–60°. JEOL JEM 1200EX and JEOL JEM 2010 transition electronic microscopy (JEOL Technics Co. Ltd., Tokyo, Japan) were used for transmission electron microscopy (TEM, JEOL Technics Co. Ltd., Tokyo, Japan) analysis and high-resolution transmission electron microscopy (HRTEM, Jeol Technics Co. Ltd., Tokyo, Japan) analysis at an accelerating voltage of 100 kV, respectively. Samples were prepared by placing one drop of an ethanol suspension of the GR/In_2_O_3_ nanocubes onto a copper grid (3 mm, 200 mesh) coated with carbon film. A JSM-7401 scanning electron microscopy (SEM, JEOL Technics Co. Ltd., Tokyo, Japan) operated at 20 kV was used to analyze the sample. The electrochemical analysis used a CHI 660D electrochemical workstation (Shanghai Chenhua Co. Ltd. Shanghai, China). Particle size distribution was tested on a laser particle size analyzer (Mastersizer 3000E, Malvern Panalytical Co. Ltd., Malvern, UK).

### 3.2. Free Amino Acid Extraction of Camellia nitidissima Chi Leaves

*Camellia nitidissima* Chi leaves were collected carefully in accordance with the Chinese Pharmacopoeia 2015 edition, then the leaves were washed and vacuum dried. In a typical procedure [13], vacuum-dried *Camellia nitidissima* Chi leaves (6.0 g) were smashed and sieved with a 50-mesh screen, then the precisely weighed sieved powder (5.0 g) was collected as the starting sample, which was transferred into a 250 mL round flask. In the next step, 70% ethanol (100 mL) was also added to the round flask, and the system was refluxed for 2 h. The mixture was cooled down and filtered, then the residual solid was re-refluxed for another 2 runs under the same conditions of the first run. All the filtrates were combined, and after bleaching, the mixture was vacuum concentrated to 80 mL. The sediment was filtered and 95% ethanol (250 mL) was added to the filtrate. The precipitate (proteins) was filtered and discarded, and the ethanol was removed using a rotary evaporator. The system was then treated with 732 cation exchange resin. The obtained compounds were washed with deionized water until the solid eluent became colorless. After this step, the solid was washed with 5% ammonia and the filtrate was collected carefully, which showed a coloration effect to ninhydrin. The filtrate was vacuum concentrated and vacuum dried at 30 °C for 8 h, resulting in a light-brown solid (337 mg), which can be used in further experiments.

### 3.3. Typical Procedures for the Preparation of GR/In_2_O_3_ Nanocubes

GO was prepared as described in previous reports [24,25]. In this work, GO dispersion (1 mg/mL in N,N-Dimethylformamide (DMF), 30 mL), indium acetylacetonate (0.21 g, 5 mol), and NaOH (0.5 g) were mixed by stirring at room temperature for 30 min. The dispersion was then sonicated for another 30 min, then the mixed solvent, EtOH-H_2_O-cyclohexane-oleic acid (30 mL, 1:0.8:1:0.4), was added. The resulting suspension was transferred into a Teflon-lined stainless-steel autoclave. The autoclave was maintained at 200 °C for 20 h and then cooled to room temperature naturally. The resulting products were filtered, washed with 0.1 mol/L HCl solution, then deionized water to pH 7, and finally vacuum dried at 65 °C for 24 h, affording the intermediate of GO/In(III). In a successive step, the prepared GO/In(III) intermediate was re-dispersed in DMF. After ultrasonication for 15 min, sodium powder (0.25 g) was added, and the mixture was then stirred at room temperature for 48 h. After the procedure, the dispersion was filtered and washed with 0.1 mol/L HCl solution and deionized water to pH 7, with a vacuum drying procedure at 65 °C for 24 h, resulting in GR/In(OH)_3_ nanocubes. Finally, the obtained intermediate was annealed at 600 °C for 2 h under Ar flow, resulting in the product: GR/In_2_O_3_ nanocubes.

### 3.4. Construction and Response Measurements of Selective Electrochemical Sensor for L-Lysine Based on GR/In_2_O_3_ Nanocubes

A GCE electrode was polished with #4, #5, #6 metallographic sandpaper sequentially to obtain a polished mirror surface. It was then polished with alumina powders of 1 and 0.3 μm, respectively. The electrode was soaked in acetone, and after taking it out, the electrode was ultrasonicated in ethanol and double-distilled water for 5 min, respectively. It was then naturally dried before use.

In a typical procedure, *Rheinheimera speciales* (RHS, representative L-lysine-ε-oxidase, 18 mg) was added to phosphate buffer (70 μL, pH 7.0) under stirring. After standing for 30 min, the upper yellow solution (25 μL) was separated and mixed with Eastman AQ-55D solution (45 μL, Sigma Aldrich, St. Louis, MO, USA). In a successive step, bovine serum protein (1.8 mg), horseradish peroxidase (HRP, 1 mg), GR/In_2_O_3_ nanocube dispersion (in DMF, 10 μL), and glutaraldehyde (12.5%, 10 μL) were added. The mixture was stirred for another 30 min, resulting in a GR/In_2_O_3_-RHS-HRP-AQ-55D mixed solution.

This mixed solution was dropped (10 μL) on the polished GCE surface, which was then dried in a refrigerator to form a film (labeled as GR/In_2_O_3_-RH-GCE electrode). A typical phosphate buffer solution (pH 7.8, 2 mL), free amino acid extraction solution of *Camellia nitidissima* Chi leaves (including extra added D-Lys with a concentration of 0.45%), or L-Lys solution (0–70 μmol·L^−1^) were added to a 10 mL electrolytic cell. The GR/In_2_O_3_-RH-GCE electrode prepared above, platinum electrode, and saturated calomel electrode (SCE) were inserted into the electrolytic cell to form a three-electrode system. The temperature was maintained at 20 °C ± 0.5, the working voltage was + 0.18V (vs. SCE), and after 200 s, the current was stabilized and the current change with the concentration of amino acids was recorded. In this work, the free amino acid extraction of *Camellia nitidissima* Chi leaves was prepared according to the method of 2.2 (previous research revealed that the extraction contained fifteen different amino acids, including L-Lys [3]).

## 4. Conclusions

In summary, we prepared novel graphene/In_2_O_3_ nanocubes successfully with a series of chemical approaches, which combined electrical properties of both graphene and In_2_O_3_ nanocubes. Detailed characterizations of the resulting nanocomposites revealed that the cubic structured In_2_O_3_ composites in a size range of 30–70 nm were evenly distributed on graphene nanosheets. In particular, L-Lys in *Camellia nitidissima* Chi can be detected readily by the GR/In_2_O_3_ nanocube-based electrochemical sensor with good chiral recognition and a broad linear range of 0.23–30 μmol·L^−1^, together with good selectivity and anti-interference properties to various other kinds of amino acids in *Camellia nitidissima* Chi. Further studies to enhance the chiral recognition, selectivity, and anti-interference properties of the GR/In_2_O_3_ nanocube-based electrochemical sensor are under progress in our laboratory.

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
