# Peer review of "Novel Graphene/In_2_O_3_ Nanocubes Preparation and Selective Electrochemical Detection for L-Lysine of *Camellia nitidissima* Chi"

_materials, 2020, doi:10.3390/ma13081999_

Round 1

Reviewer 1 Report

The composite material GR/Ir2O3-cubes is interesting and the resulting enantioselective sensor for L-Lys seems to perform well. However, the work presented is minimalistic (i.e. reduced to the minimum essential) lacking necessary testing of reproducibility in sensing, quantitative measurement of chiral selectivity, and selectivity from other amino acids, among those listed in Line 118 only D-Lys and L-Asp have been tested with only one addition each. The authors should do more work to complete their experimental results including reproducibility of the sensor (would three or more sensors have the same response?), provide signal-to-noise ratios from chronoamperometric curve across the linear range, and complete the study of selectivity against D-Lys and L-Asp over relevant ranges of concentration.

The English form, especially the use of the past tense, should be revised through the manuscript, which reads awkwardly. For example, "The research on the homochirality of basic biological molecules such as amino acids WAS still an intriguing puzzle in mordern [sic] sciences" where the use of "WAS" does not say if it is now an intriguing puzzle, so does it matter to study this now? Line 40: "had focused" should be "has focused" otherwise are the reports still timely for this study?

Abstract:

Line 9: Remove "a" from "a novel".

Line 12: Change "resulted" to "resulting".

Line 19: Change "linearly" to "linear".

Introduction:

The Introduction is very short, and the last paragraph is the same of abstract. The authors should not repeat the abstract and the introduction should be expanded.

Line 30: provide relevant details of the reports on transition-metal nanocubes/graphene nanocomposites since the authors state that there are a few, how are these different and valuable for the study presented by the authors?

Line 36: Expand this paragraph. What is the importance of Camellia nitidissima Chi? Is it a source of amino acids? How is it used? What role does it have in the Chinese and possibly global food industry and economy? Etc.

Lines 42-49: This section is the same of the abstract, as such should be removed or changed. The conclusions are also a repetition of the abstract.

Scheme 1 and relevant text in the manuscript: Why is In3+ not reduced to In0 in the presence of metallic sodium Na?

Figure 1: The scale of a) and c) cannot be seen, please add them.

Figure 3: A and B should be lower case to be consistent with previous figures. Why is there copper in the EDX? What additional information is obtained from the EDX spectrum? Mapping of composites is more interesting usually to see elements distribution.

Line 106: Please specify what is meant by "etc."

Scheme 2: Please add a better quality image for CHI cyclic voltammetry.

Lines 113-122: Where are the experimental results in support of this section? Only the anodic peak current response for L-Lys is reported in Figure 5. D-Lys and L-Asp are in Figure 6, but a single test with a single concentration is not enough to demonstrate the performance of the sensor. What concentration range was tested for D-Lys? Also, since L-Asp is oxidised at the same potential of L-Lys, what is the max concentration of L-Asp that do not interfere with the detection limit of L-Lys? How reproducible is the preparation of the sensor? What is the signal-to-noise ratio in the chronoamperometric curves?

Experimental

Line 179: The potentiostat might be CHI 660D not CHI 6660D.

Line 188: Please specify if the other two runs were performed as the first one.

Line 190: The term "precipitation" should be "precipitate". Also, it is unclear what the authors kept at this stage if the precipitate was discarded and the filtrate was rotovaped: what was left? What solid became colourless? Please clarify.

Line 198: "was mixed" should be "were mixed".

Author Response

                           Responses to Reviewers

-------------------------------------------------------------

Response to Reviewer #1

-------------------------------------------------------------

General comments

The composite material GR/Ir2O3-cubes is interesting and the resulting enantioselective sensor for L-Lys seems to perform well. However, the work presented is minimalistic (i.e. reduced to the minimum essential) lacking necessary testing of reproducibility in sensing, quantitative measurement of chiral selectivity, and selectivity from other amino acids, among those listed in Line 118 only D-Lys and L-Asp have been tested with only one addition each. The authors should do more work to complete their experimental results including reproducibility of the sensor (would three or more sensors have the same response?), provide signal-to-noise ratios from chronoamperometric curve across the linear range, and complete the study of selectivity against D-Lys and L-Asp over relevant ranges of concentration.

Author reply: We thank the reviewer for his/her recognition of our efforts and important comments. All the comments have been addressed in the revised version and replied point by point in the response letter. Further work on reproducibility in sensing, quantitative measurement of chiral selectivity were also added in the revised mauscript (for example, expanded introduction section, new conducted work showed in Fig. 5b and Fig. 7, as well as a series of re-test results, please see the revised manuscript, which were marked red). We believe that the quality of the revised manuscript has been greatly improved.

Detailed comments

  1. The English form, especially the use of the past tense, should be revised through the manuscript, which reads awkwardly. For example, "The research on the homochirality of basic biological molecules such as amino acids WAS still an intriguing puzzle in mordern [sic] sciences" where the use of "WAS" does not say if it is now an intriguing puzzle, so does it matter to study this now? Line 40: "had focused" should be "has focused" otherwise are the reports still timely for this study?

Author reply: Many thanks for the reviewer for pointing out our unsuitable expression. In the revised manuscript, we had checked the English form carefully, especially usage of the tense.

For example, “.... basic biological molecules such as amino acids was still an intriguing puzzle in mordern sciences”( lines 37-38 in the original manuscript) was changed to “.... basic biological molecules such as amino acids is still an intriguing puzzle in mordern sciences”. Line 40: “However, until now, few reports had focused on....”: was changed to “However, until now, few reports has focused on...”(for detail, please see the revised manuscript).

  1. Abstract:     Line 9: Remove "a" from "a novel".  Line 12: Change "resulted" to "resulting".  Line 19: Change "linearly" to "linear".

Author reply: We thank the reviewer for raising this important concern. In this work, all the wrong expressions were corrected, for example, Line 9: “In this work, a novel...” was changed to “In this work, novel....”; Line 12:“The resulted nanocomposites ...” was changed to “The resulting nanocomposites ....”.   Line 19: “...with a linearly range...” was changed to “...with a linear range.......”(for detail, please see the revised manuscript).

  1. Introduction:  The Introduction is very short, and the last paragraph is the same of abstract. The authors should not repeat the abstract and the introduction should be expanded.

Author reply: We thank the reviewer for raising this important concern. In this revised manuscript, expanded expression was added in the introduction section. The last paragraph was rewritten carefully to avoid repetition with the abstract section (showed below as well as the revised manuscript).

  Introduction section:

In the past years, graphene has gained various of interests in different fields1-5. This novel 2D carbon-based nanomaterial possesses superior chemical, physical and electrical properities6-10. A variety of intriguingly graphene based nanocomposites with different morphologies, for instance, nanoparticles11, nanoshuttles3, nanorods12, nanofibers13, nanosheet14, metal organic frameworks (MOFs)15 and quantum dots16 etc., were prepared with different applications. Recently, some researchers studied varous functional graphene nanocomposites based electrochemical sensor for different applications. For exmaple, our team developed sodium dodecyl benzene sulphonate (SDBS) functionalized graphene sheets based electrochemical biosensor, which showed excellent electrocatalytic performance toward the reduction of H2O2 with fast response, wide linear range, high sensitivity and good stability17. Kang et al. employed thermally split graphene oxide, both of which exhibit similar excellent direct electrochemistry of glucose oxidase (GOD)18. However, there were few reports concerning synthesis and application of transition-metal nanocubes/graphene nanocomposites.

Camellia nitidissima Chi, a beautiful widely used plant in southern China, has good reputation of "Giant Panda of Botany"3. In 2010, the Ministry of Health of the People's Republic of China approved Camellia nitidissima Chi as one kind of the new resource food (2010 Chinese Ministry of Health Announcement No. 9). Investigation revealed that Camellia nitidissima Chi contains rich amino acids, vitamin C, gross sugar and protein etc. Moreover, the contents of some typical essential amino acids (EAAs) in cultivated Camellia nitidissima Chi, for example, Leu, Thr, Val and Ile, were obvious higher than Food and Agriculture Organization of the United Nations (FAO)/World Health Organization (WHO) refence contents19. Our previous research also showed that by graphene nanoshuttles matrix based matrix-assisted laser desorption/ionization time of flight mass spectrometry(MALDI-TOF MS), fifteen kinds of amino acids ingredients, including L-lysine (L-Lys), L-aspartic acid (L-Asp), L-threonine (L-Thr), L-serine (L-Ser) and L-glutamic acid (L-Gly) etc., can be detected readily in Camellia nitidissima Chi with strong peaks and good discrimination3. Similar results were also observed in our another work. By combing with gas chromatography-mass spectrometry (GC-MS), the combined stir bar sorptive extraction (SBSE)/GC-MS technology with linear graphene nanocomposites coating can detect seventeen kinds of amino acids of Camellia nitidissima Chi seeds, including Ala, Gly, Thr, Ser, Val, Leu, Ile, Cys, Pro, Met, Asp, Phe, Glu, Lys, Tyr, His and Arg 13.

The research on the homochirality of basic biological molecules such as amino acids is still an intriguing puzzle in mordern sciences20. Research on chiral recognition and chiral selection of some kinds of amino acids21 could provide fundamental scientific thought in resolving the puzzle of so kind homochirality and other important applications. However, until now, few reports has focused on chiral recognition of single kind of amino acid, for example, L-Lys, in Camellia nitidissima Chi. 

In this study, by one-pot solvothermal treatment together with reduction and annealing treatments in turn, we successfully prepared novel graphene/In2O3 (GR/In2O3) nanocubes. The prepared nanocomposites were detailed characterized by transmission electron microscopy, scanning electron microscopy and X-ray diffraction spectroscopy etc. A selective electrochemical sensors for L-Lys of Camellia nitidissima Chi based on GR/In2O3 nanocubes was designed and applied. In linear range of 0.23-30μmol·L-1, the GR/In2O3 based electrochemical sensors illustrated good chiral recognition to L-Lys of Camellia Nitidissima Chi with strong selective signal.

  1. Line 30: provide relevant details of the reports on transition-metal nanocubes/graphene nanocomposites since the authors state that there are a few, how are these different and valuable for the study presented by the authors?

Author reply: We thank the reviewer for your comments. However, we should emphasized that we expressed “there were few reports concerning ....transition-metal nanocubes...” in this manucript rather than “there were a few reports concerning ....transition-metal nanocubes...”.

  1. Line 36: Expand this paragraph. What is the importance of Camellia nitidissima Chi? Is it a source of amino acids? How is it used? What role does it have in the Chinese and possibly global food industry and economy? Etc..

Author reply: We thank the reviewer for raising this important concern. In the revised manuscript, we had expanded the paragraph concerning the importance of Camellia nitidissima Chi (showed below, also showed in the revised manuscript):

 Camellia nitidissima Chi, a beautiful widely used plant in southern China, has good reputation of "Giant Panda of Botany"3. In 2010, the Ministry of Health of the People's Republic of China approved Camellia nitidissima Chi as one kind of the new resource food (2010 Chinese Ministry of Health Announcement No. 9). Investigation revealed that Camellia nitidissima Chi contains rich amino acids, vitamin C, gross sugar and protein etc. Moreover, the contents of some typical essential amino acids (EAAs) in cultivated Camellia nitidissima Chi, for example, Leu, Thr, Val and Ile, were obvious higher than Food and Agriculture Organization of the United Nations (FAO)/World Health Organization (WHO) refence contents19. Our previous research also showed that by graphene nanoshuttles matrix based matrix-assisted laser desorption/ionization time of flight mass spectrometry(MALDI-TOF MS), fifteen kinds of amino acids ingredients, including L-lysine (L-Lys), L-aspartic acid (L-Asp), L-threonine (L-Thr), L-serine (L-Ser) and L-glutamic acid (L-Gly) etc., can be detected readily in Camellia nitidissima Chi with strong peaks and good discrimination3. Similar results were also observed in our another work. By combing with gas chromatography-mass spectrometry (GC-MS), the combined stir bar sorptive extraction (SBSE)/GC-MS technology with linear graphene nanocomposites coating can detect seventeen kinds of amino acids of Camellia nitidissima Chi seeds, including Ala, Gly, Thr, Ser, Val, Leu, Ile, Cys, Pro, Met, Asp, Phe, Glu, Lys, Tyr, His and Arg 13.

  1. Lines 42-49: This section is the same of the abstract, as such should be removed or changed. The conclusions are also a repetition of the abstract.

Author reply: We thank the reviewer for raising this important concern. In the revised manuscript, both the last paragraph in the introduction section and the conclusion section were revised according to the valuable comments of the reviewer (also showed belowed).

  The last paragraph in the introduction section:

In this study, by one-pot solvothermal treatment together with reduction and annealing treatments in turn, we successfully prepared novel graphene/In2O3 (GR/In2O3) nanocubes. The prepared nanocomposites were detailed characterized by transmission electron microscopy, scanning electron microscopy and X-ray diffraction spectroscopy etc. A selective electrochemical sensors for L-Lys of Camellia nitidissima Chi based on GR/In2O3 nanocubes was designed and applied. In linear range of 0.23-30μmol·L-1, the GR/In2O3 based electrochemical sensors illustrated good chiral recognition to L-Lys of Camellia Nitidissima Chi with strong selective signal.

The conclusion section:

In summary, we prepared novel graphene/In2O3 nanocubes successfully by a series of chmical approaches, which can combine both electrical properties of graphene and In2O3 nanocubes. Detailed characterizations of the resulted nanocomposites revealed that the cubic structured In2O3 composites in a size range of 30-70 nm were evenly distributed on graphene nanosheets. In particular, L-Lys in Camellia Nitidissima Chi can be detected reaily by the GR/In2O3 nanocubes based electrochemical sensor with good chiral recognition and broad linear range of 0.23-30μmol·L-1, together with good selectivity and anti-interference properties to various other kinds of amino acids in Camellia Nitidissima Chi. Further study to enhance the chiral recognition, selectivity and anti-interference properties of the GR/In2O3 nanocubes based electrochemical sensor are under progress in ou laboratory.

  1. Scheme 1 and relevant text in the manuscript: Why is In3+ not reduced to In0 in the presence of metallic sodium Na?

Author reply: We thank the reviewer for raising this important concern. As shown in modified Scheme 1 in the revised manuscript: In the first step, no any sodium Na was added, at room temperature, In3+ could not be reduced to In0 (Na powder was added after this step). We also check the text, we had not mentioned that In3+ can be reduced to In0 without addition of Na powder.

Scheme 1. Synthesis of GR/In2O3 nanocubes.

  1. Figure 1: The scale of a) and c) cannot be seen, please add them.

Author reply: We thank the reviewer for pointing out our mistakes. We are sorry for missing the reference size in Fig. 1a, together with unclear size scale in Fig. 1b (not Fig. 1c), in the revised manuscript, we are glad to provide clear reference size in both Fig. 1a and Fig. 1b (the revision showed below).

Meanwhile, as shown in Fig. 1c, it has clear scale.

Fig. 1. a-b)TEM of GR/In2O3 nanocubes, c-d) HRTEM of GR/In2O3 nanocubes, inseted image in Fig 1c was electron diffraction of the prepared GR/In2O3 nanocubes.

  1. Figure 3: A and B should be lower case to be consistent with previous figures. Why is there copper in the EDX? What additional information is obtained from the EDX spectrum? Mapping of composites is more interesting usually to see elements distribution.

Author reply: We thank the reviewer for raising this important concern. Sorry for our misunderstanding expression, in the revised manuscript, we had changed EDX to the EDS (Energy Dispersive X-Ray Spectroscopy), which used copper mesh (see below image) to support the sample. We obtained the EDS data after TEM observation on copper mesh supported sample. Therefore, peaks of copper element appeared in the obtained EDS image.

   Thank you very much for reviewer’s kind guidance. Yes, mapping of composites is more interesting usually to see elements distribution, we are glad to try this method in the further research.

Meanwhile, EDS analysis was also important method for nanomaterials characterization, for example, Bidabadi A S, Korinek A, Botton G A, et al. High resolution transmission electron microscopy (TEM), energy-dispersive X-ray spectroscopy (EDS) and X-ray diffraction studies of nanocrystalline manganese borohydride (Mn (BH4) 2) after mechano-chemical synthesis and thermal dehydrogenation[J]. Acta materialia, 2015, 100: 392-400; Patri A, Umbreit T, Zheng J, et al. Energy dispersive X‐ray analysis of titanium dioxide nanoparticle distribution after intravenous and subcutaneous injection in mice[J]. Journal of Applied Toxicology: An International Journal, 2009, 29(8): 662-672; Lu P, Zhou L, Kramer M J, et al. Atomic-scale chemical imaging and quantification of metallic alloy structures by energy-dispersive X-ray spectroscopy[J]. Scientific reports, 2014, 4(1): 1-5.

  1. Line 106: Please specify what is meant by "etc."

Author reply: We thank the reviewer for raising this important concern. In the revised manuscript, as described in section of 3.4, besides Rheinheimera sp. and horseradish peroxidase, “etc.” meant Eastman AQ-55D solution and bovine serum protein and glutaraldehyde etc. For details, please see below description (3.4 in the revised manuscript):

In a typical procedures, Rheinheimera sp. (RHS, representative L-lysine-ε-oxidase, 18 mg) was added to phosphate buffer (70 μl, pH 7.0) under stirring. After standing for 30min, the upper yellow solution(25 μl) was separated and mixed with Eastman AQ-55D solution (45μl ). In a successive step, bovine serum protein (1.8 mg ), horseradish peroxidase (HRP, 1 mg), GR/In2O3 nanocubes dispersion (in DMF, 10 μl ) and glutaraldehyde (12.5%, 10μl) was added, the mixture was stirred for another 30 mins, giving GR/In2O3-RHS-HRP-AQ-55D mixed solution.

  1. Scheme 2: Please add a better quality image for CHI cyclic voltammetry.

Author reply: We thank the reviewer for your valuable comments. In this revised manuscript, we had added a better quality image for CHI cyclic voltammetry. Thank you very much for your kind guidance (also showed below):

Scheme 2. Schematic diagram of GR/In2O3 nanocubes based selective electrochemical sensor for L-Lysine of Camellia Nitidissima Chi

  1. Lines 113-122: Where are the experimental results in support of this section? Only the anodic peak current response for L-Lys is reported in Figure 5. D-Lys and L-Asp are in Figure 6, but a single test with a single concentration is not enough to demonstrate the performance of the sensor. What concentration range was tested for D-Lys? Also, since L-Asp is oxidised at the same potential of L-Lys, what is the max concentration of L-Asp that do not interfere with the detection limit of L-Lys? How reproducible is the preparation of the sensor? What is the signal-to-noise ratio in the chronoamperometric curves?

Author reply: We thank the reviewer for raising this important concern. In the revised manuscript, we had re-tested the response curves for the current of L-Lys and D-Lys by the prepared GR/In2O3 nanocubes based electrochemical sensor (illustrated in Fig. 5 of the revised manuscript), in which, the concentration range was also tested for D-Lys.

For L-Arg (not L-Asp), according to previous report of “ Materials 10, 443 (2017)”, L-Arg and L-Lys had contiguous contents (L-Arg 5.0mg/g, L-Lys 5.1mg/g) in Camellia Nitidissima Chi, together with similar structures, therefore, feeble response signal of L-Arg was detected in this work. Owing to relatively fixed contents of L-Arg and L-Lysin in Camellia Nitidissima Chi, it’s difficulty to eliminate the interfere of L-Arg when detecting L-Lys in Camellia Nitidissima Chi.

   We appreciate the reviewer’s valuable comments, recently, we carried our a series of extra experiments at different L-Arg concentrations and L-Arg (L-Lys was added at a concentration of 0.45%, both L-Arg and L-Lys were standard substances), once the concentration of L-Arg was controlled below 0.31% (<0.31%), it would not  interfere with the detection of L-Lys. No L-Arg signal can be detected.

   For reproducible is the preparation of the sensor and other valuable questions of the reviewer’s, as we can see from Fig. 5 and Fig. 7 of the revised manuscript, with a broad linear range of 0.23-30μmol·L-1, differential pulse voltammetry (DPV) responses of L-Lys by GR/In2O3 nanocubes based electrochemical sensor were also detected. As shown in Fig. 7, after using the GR/In2O3 nanocubes based electrochemical sensor, the current response values of the L-Lys on the electrodes were significantly negatively shifted (as shown in Fig. 7a, when electrode without GR/In2O3 nanocubes was used, the value was 0.84V, while as illustrated in Fig. 7b, the oxidation peak potential of the modified electrode changed to 0.72V together with much improved peak intensity), which indicates that after modified with GR/In2O3 nanocubes, the adsorption amount and response speed of L-Lyse were significantly improved. Meanwhile, the signal-to-noise ratio in the chronoamperometric curves was 3.

Fig. 7 Differential pulse voltammetry (DPV) responses of L-Lys by (a) electrode without GR/In2O3 nanocubes; (b) GR/In2O3 nanocubes based electrochemical sensor.

  1. Experimental  Line 179: The potentiostat might be CHI 660D not CHI 6660D.

Author reply: We appreciate the reviewer very much for pointing out our mistake. In the revised manuscript, we had corrected “CHI 6660D” to “CHI 660D”. Thanks again for your kind help. 

  1. Line 188: Please specify if the other two runs were performed as the first one.

Author reply: We thank the reviewer for raising this important concern. In the revised manuscript, the corresponding expression at line 188 was changed to: “The mixture was cooled down and filtered, then the residual solid was re-refluxed for another 2 runs under the same conditions of the first run.”(please see the revised manuscript).

  1. Line 190: The term "precipitation" should be "precipitate". Also, it is unclear what the authors kept at this stage if the precipitate was discarded and the filtrate was rotovaped: what was left? What solid became colourless? Please clarify.

Author reply: We thank the reviewer for valuable comments. In the revised manuscript, the term "precipitation" was corrected to "precipitate".

Meanwhile, in the “3.2 Free amino acids extraction of Camellia nitidissima Chi Leaves” section (lines 151-195), the method of which was in accordance with literature reports, for example, Chen, Z.; Liu, Q.; Geng, Z.; Tang, G. Extraction- separation and determination of amino acid in seeds of Rubia Cordifolia L. J. Shanxi Norm. Univ. (Nat. Sci. Ed.) 1996, 12, 118–119 and Cheng J., Zhong R., Lin J. et al. Linear graphene nanocomposite synthesis and an analytical application for the amino acid detection of Camellia nitidissima Chi seeds. Materials, 2017, 10, 443. 

In such typical procedures, the precipitate (proteins) was filtered and discarded, the ethanol extraction, which containing crude amino acids and other impurity substances, was treated with rotary evaporator, under which, the ethanol was removed, giving residual crude amino acids and other impurity substances.

Lines 191-192: “The obtained 192 compounds were washed with deionized water until the solid became colorless.” We were really sorry for our mistake, in the 1st-manuscript, we had missed the word “eluent”. In the revised manuscript, we had corrected the expression showed below: “The obtained 192 compounds were washed with deionized water until the solid eluent became colorless.” Thank you very much for the reviewer’s key-important comments.

  1. Line 198: "was mixed" should be "were mixed".

Author reply: We thank the reviewer for for pointing out our mistake. In the revised manuscript, we had corrected “was mixed” to “were mixed”. Thanks again for your kind help.

Reviewer 2 Report

In this article the Authors describe the synthesis of an electrochemical, selective sensor for L-Lysine. The sensor was prepared by one-pot solvothermal procedure, by reacting graphene oxide and indium acetylacetonate. The sensor structure was charcterized by means of TEM, SEM and XRD. The performance and the selctivity of the sensor was evaluated, showing a linear recognition in the concentration range of 0.23-30μmol·L-1. The selectivity towards L-Arginine and D-Lysine was also investigated.

In my opinion the drafting of this article is sufficiently adequate, however it needs to be improved in some aspects before it can be accepted for publication in this journal.

  1. The introduction should be improved, higlighting the novelty of the work and citing more relevant references in the field.
  2. Figure 1 a and b: the reference size are missing.
  3. Figure 2: which is the magnification? Are better images available?
  4. Figure 3B: translate the Chinese and change the font and size of the scale
  5. Line 72: how the size was calculated or measured the size range?
  6. Line 240: etc..... Which are the other analysis?

Author Response

                        Responses to Reviewers

-------------------------------------------------------------

Response to Reviewer #2

-------------------------------------------------------------

General comments

In this article the Authors describe the synthesis of an electrochemical, selective sensor for L-Lysine. The sensor was prepared by one-pot solvothermal procedure, by reacting graphene oxide and indium acetylacetonate. The sensor structure was charcterized by means of TEM, SEM and XRD. The performance and the selctivity of the sensor was evaluated, showing a linear recognition in the concentration range of 0.23-30μmol·L-1. The selectivity towards L-Arginine and D-Lysine was also investigated. In my opinion the drafting of this article is sufficiently adequate, however it needs to be improved in some aspects before it can be accepted for publication in this journal.  

Author reply: Thank you very much for the reviewer’s positive comments. In this work, we are glad to report the GR/In2O3 nanocubes based electrochemical sensor fabrication and application, which showed good chiral recognition feature to L-lysine of Camellia Nitidissima Chi with a linearly range of 0.23-30μmol·L-1, together with selectivity and anti-interference properties to other kinds of amino acids of Camellia Nitidissima Chi. We appreciate the valuable suggestions from you very much. In the revised manuscript, we addressed each of your valuable comments. Our responses to the comments are list below point by point followed by each specific reply.

Detailed comments

  1. The introduction should be improved, higlighting the novelty of the work and citing more relevant references in the field.

Author reply: Many thanks for the valuable comments of the reviewer. In the revised mauscript, the introduction section was rewritten to higlight the novelty of the work and cite more relevant references in the field, according to the important comments of the reviewer’s (please see the revised manuscript).

  1. Figure 1 a and b: the reference size are missing.

Author reply: We thank the reviewer for raising this important concern. We are sorry for missing the reference size in Fig. 1a, together with unclear size scale in Fig. 1b, in the revised manuscript, we are glad to provide clear reference size in both Fig. 1a and Fig. 1b (the revision showed below):

Fig. 1. a-b)TEM of GR/In2O3 nanocubes, c-d) HRTEM of GR/In2O3 nanocubes, inseted image in Fig 1c was electron diffraction of the prepared GR/In2O3 nanocubes.

  1. Figure 2: which is the magnification? Are better images available?

Author reply: Dear reviewer, we appreciate your valuable suggestions. In the revised manuscript, we replaced the original Fig. 2a-b with new SEM image (please see Fig. 2 in the revised manuscript, also showed below), the magnification of which was ×50000. Due to high magnification was displayed in Fig. 2 of the revised manuscript, the image resolution decreased slightly, thank you for your kind understanding.

 Fig. 2 SEM images of GR/In2O3 nanocubes.

  1. Figure 3B: translate the Chinese and change the font and size of the scale

Author reply: Dear reviewer, thanks a lot for pointing these out. In the revised manuscript, we had translate the Chinese expression and change the font and size of the scale of Fig. 3B (also showed below).

Fig. 3 (A) XRD spectra of a)graphene, b)GR/In(OH)3 intermediate and c) GR/In2O3 nanocubes, (B) EDS analysis of GR/In2O3 nanocubes.

  1. Line 72: how the size was calculated or measured the size range?

Author reply: We thank the reviewer for his/her important comments. During the TEM and SEM measurement processes, we scaled the size range of the synthesized GR/In2O3 nanocubes. There were altogether 17 pieces of GR/In2O3 nanocubes were scaled, all of which were among the size range of 30-70 nm. We can also observe the approximate size range of the nanocubes in Fig. 1-2.

  1. Line 240: etc..... Which are the other analysis?

Author reply: Many thanks for the reviewer’s important comment. Besides  transmission electron microscopy (Fig. 1a-b, d), scanning electron microscopy (Fig. 2) and X-ray diffraction spectroscopy (Fig. 3A) technologies was used to characterized the synthesized GR/In2O3 nanocubes, other analysis methods, for example, high-resolution transmission electron microscopy (HRTEM) (Fig. 1c) and energy dispersive X-ray analysis (EDS, Fig. 3B) were further used to observe the surface morphology of the prepared nanocomposites.

Reviewer 3 Report

This manuscript introduces graphene/In2O3 nanocubes for biosensor application. I think this paper is worth publishing in this journal. The following are minor comments;

1) Please carefully check the typos. (For example, in Pages 2 & 10, naocube -> nanocube)

2) Figure 2 should be taken again. The resolution is too poor.

3) It is better to display the peak information in Figure 3b.

4) Please discuss the size uniformity of nanocubes. Can you quantitatively show the size distribution of a nanocube? How does this size non-uniformity affect sensor performance?

Author Response

                    Responses to Reviewers

-------------------------------------------------------------

Response to Reviewer #3

-------------------------------------------------------------

General comments

This manuscript introduces graphene/In2O3 nanocubes for biosensor application. I think this paper is worth publishing in this journal. The following are minor comments.  

Author reply: Thank you very much for the reviewer’s positive comments. In the revised manuscript, we addressed each of your valuable comments. Our responses to the comments are list below point by point followed by each specific reply.

Detailed comments

  1. Please carefully check the typos. (For example, in Pages 2 & 10, naocube -> nanocube)

Author reply: Many thanks for the valuable comments of the reviewer. In the revised manuscript, we had checked the typos carefully. In pages 2 and 10, the wrong expression “naocube” was corrected to “nanocube” (please see the revised manuscript, all changes were marked red).

  1. Figure 2 should be taken again. The resolution is too poor.

Author reply: We thank the reviewer for raising this important concern. In the revised manuscript, Fig. 2a with improved resolution was taken again (which was also showed below):

                                        Fig. 2

  1. It is better to display the peak information in Figure 3b.

Author reply: Dear reviewer, we appreciate your valuable suggestions. In the revised manuscript, We had added peak information in curve b of Figure 3A (also showed below):

The peaks at 2θ=22.26°, 31.68°, 39.08°, 45.42°, 51.56° and 56.48° can be assigned to (200), (220), (222), (400), (420) and (422) crystalline plane diffraction peaks of In(OH)3 (JCPDS no. 85-1338, curve b in Fig. 2A). 

  1. Please discuss the size uniformity of nanocubes. Can you quantitatively show the size distribution of a nanocube? How does this size non-uniformity affect sensor performance?

Author reply: Dear reviewer, thanks a lot for pointing these out. In the revised manuscript, we had quantitatively showed the size distribution of a nanocube in Fig. 1Sb (also showed below, tested on a Laser particle size analyzer of Mastersizer 3000E), which was in a size range of 30-70nm.

                                             Fig. 1S

 Meanwhile, As described in 3.3 of experimental section, “in a successive step, the prepared GO/In(III) intermediate was re-dispersed in DMF, after ultrasonication for 15min, sodium powder (0.25g) was added, the mixture was then stirred under room temperature for 48h. After the procedure, the dispersion was filtered and washed with 0.1 mol/L HCl solution and deionized water to pH 7, with vacuum drying procedure at 65 oC for 24 h, giving GR/In(OH)3 nanocubes. Finally, the obtained intermediate was annealing at 600 oC for 2h under Ar flow, giving the product: GR/In2O3 nanocubes”. Recently, we had conducted modified preparation procedures. In such procedures, we adjust the amount of sodium powder addition to 0.15g, stirring time was changed to 24h, the annealing conditions were modified to: 550 oC for 1.5h under Ar flow. Under such conditions, GR/In2O3 nanocubes with smaller size range of 20-50 nm was obtained. The obtained GR/In2O3 nanocubes (20-50 nm) was used to prepare electrochemical sensor of amino acids extraction for Camellia Nitidissima Chi under the same conditions, which showed inferior performance comparing with the results showed in Fig. 4 (with size range of 30-70 nm) of the revised manuscript. As illustrated in Fig. 3S, besides peaks of L-Lys, D-Lys and L-Arg, interferential peak of L-Asp was also observed, together with obvious stronger interference peak of L-Arg and D-Lys in Fig. 3S.

                                                       Fig. 4

                                                      Fig. 3S

Round 2

Reviewer 1 Report

The authors have partially worked to address my comments since there are still some concerns - a major one in particular that undermines the central claim of the study.

Major:

The new data provided in Figure 5 show that the sensor is not that enantioselective as initially and currently claimed by the authors, this is a severe aspect, and I am wondering why this was not tested/disclosed originally by the authors? At amino acid concentration below 10 um/L the linear responses for L-Lys and D-Lys are very similar. In order to establish whether there is any chiral selectivity in this range, error bars associated with the reproducibility of the sensor response should be added. This is the minimum expectation for sensor characterisation. At least six separate sensors should be made; three tested to detect L-Lys and three tested for D-Lys. These experiments will produce at least six response curves like in Figure 5, the average curve with standard deviations for each data point for L-Lys will be calculated from the three response curves for L-Lys.

Similarly, the average curve with standard deviations for each point for D-Lys will be calculated from the three response curves for D-Lys. The two average curves with their corresponding standard deviations (error bars) will be then compared in a new plot to be used in place of the one given in Figure 5. If the error bars of the average response curve overlap, there is no chiral selectivity in the low concentration range, which is most relevant for the operation of a sensor. This experimental work is critical since it underpins the central claim of the study. Is the sensor really enantioselective? Without this minimal set of measurements, the authors cannot answer this question and support their claim.

Other minor points:

Regarding comment 7: Based on Scheme 1, Na metal is added to the composite including Ir(OH)3, please specify in the paper why would Ir(III) not be reduced to Ir(0).

The quality of Figure 2 is quite poor. Would it be worth putting a better one? Or shrink it to make it look better?

Also:

Please make sure to check grammar and spelling in the next revision, "For exmaple" on page 1, for example.

Author Response

                              Responses to Reviewers

-------------------------------------------------------------

Response to Reviewer #1

-------------------------------------------------------------

General comments

The authors have partially worked to address my comments since there are still some concerns - a major one in particular that undermines the central claim of the study.

Author reply: We thank the reviewer for his/her important comments. All the important comments by the reviewer 1 have been addressed in the revised version and replied point by point in the response letter (please see the revised manuscript, which were marked red). We believe that the quality of the revised manuscript has been greatly improved.

Detailed comments

  1. The new data provided in Figure 5 show that the sensor is not that enantioselective as initially and currently claimed by the authors, this is a severe aspect, and I am wondering why this was not tested/disclosed originally by the authors? At amino acid concentration below 10 um/L the linear responses for L-Lys and D-Lys are very similar. In order to establish whether there is any chiral selectivity in this range, error bars associated with the reproducibility of the sensor response should be added. This is the minimum expectation for sensor characterisation. At least six separate sensors should be made; three tested to detect L-Lys and three tested for D-Lys. These experiments will produce at least six response curves like in Figure 5, the average curve with standard deviations for each data point for L-Lys will be calculated from the three response curves for L-Lys. Similarly, the average curve with standard deviations for each point for D-Lys will be calculated from the three response curves for D-Lys. The two average curves with their corresponding standard deviations (error bars) will be then compared in a new plot to be used in place of the one given in Figure 5. If the error bars of the average response curve overlap, there is no chiral selectivity in the low concentration range, which is most relevant for the operation of a sensor. This experimental work is critical since it underpins the central claim of the study. Is the sensor really enantioselective? Without this minimal set of measurements, the authors cannot answer this question and support their claim.   

Author reply: We thank the reviewer for raising this important concern. Your original comments for Fig. 5 were showed below, which was revised carefully according to your kind suggestion. We are very puzzled why the reviewer 1had so many critical questions. In the first run, we had answered totally 16 pieces of comments by reviewer 1 point by point carefully.We were also wondering if there are serious conflict interests between reviewer 1 and us.

  By the way, the mentioned unit of “10 um/L” by reviewer 1 was wrong.

Orginal comments by reviewer 1 in the 1st run:

Lines 113-122: Where are the experimental results in support of this section? Only the anodic peak current response for L-Lys is reported in Figure 5. D-Lys and L-Asp are in Figure 6, but a single test with a single concentration is not enough to demonstrate the performance of the sensor. What concentration range was tested for D-Lys? Also, since L-Asp is oxidised at the same potential of L-Lys, what is the max concentration of L-Asp that do not interfere with the detection limit of L-Lys? How reproducible is the preparation of the sensor? What is the signal-to-noise ratio in the chronoamperometric curves?

Author reply (1st run): We thank the reviewer for raising this important concern. In the revised manuscript, we had re-tested the response curves for the current of L-Lys and D-Lys by the prepared GR/In2O3 nanocubes based electrochemical sensor (illustrated in Fig. 5 of the revised manuscript), in which, the concentration range was also tested for D-Lys.

  1. Regarding comment 7: Based on Scheme 1, Na metal is added to the composite including Ir(OH)3, please specify in the paper why would Ir(III) not be reduced to Ir(0).

Author reply: We thank the reviewer for raising this important concern. In the first run, we had answered similar questions.

   Most important, we must emphasized that the intermediate In(OH)3 was prepared  instead of Ir(OH)3, In(III) cation was used instead of Ir(III) cation.

   Similar mistakes were happened in the 1st run by the reviewer 1, what happened ? Why?

Orginal comments by reviewer 1 in the 1st run:

  The composite material GR/Ir2O3-cubes is.....

Scheme 1 and relevant text in the manuscript: Why is In3+ not reduced to In0 in the presence of metallic sodium Na?

Author reply in the 1st run: We thank the reviewer for raising this important concern. As shown in modified Scheme 1 in the revised manuscript: In the first step, no any sodium Na was added, at room temperature, In3+ could not be reduced to In0 (Na powder was added after this step). We also check the text, we had not mentioned that In3+ can be reduced to In0 without addition of Na powder.

               Scheme 1. Synthesis of GR/In2O3 nanocubes.

     There were also other mistakes by reviewer 1 in the 1st run, for example:

  • Line 30: provide relevant details of the reports on transition-metal nanocubes/graphene nanocomposites since the authors state that there are a few, how are these different and valuable for the study presented by the authors?

Author reply: We thank the reviewer for your comments. However, we should emphasized that we expressed “there were few reports concerning ....transition-metal nanocubes...” in this manucript rather than “there were a few reports concerning ....transition-metal nanocubes...”.

  • Figure 3: A and B should be lower case to be consistent with previous figures. Why is there copper in the EDX? What additional information is obtained from the EDX spectrum? Mapping of composites is more interesting usually to see elements distribution.

Author reply: We thank the reviewer for raising this important concern. Sorry for our misunderstanding expression, in the revised manuscript, we had changed EDX to the EDS (Energy Dispersive X-Ray Spectroscopy), which used copper mesh (see below image) to support the sample. We obtained the EDS data after TEM observation on copper mesh supported sample. Therefore, peaks of copper element appeared in the obtained EDS image.

   Thank you very much for reviewer’s kind guidance. Yes, mapping of composites is more interesting usually to see elements distribution, we are glad to try this method in the further research. Meanwhile, EDS analysis was also important method for nanomaterials characterization, for example, Bidabadi A S, Korinek A, Botton G A, et al. High resolution transmission electron microscopy (TEM), energy-dispersive X-ray spectroscopy (EDS) and X-ray diffraction studies of nanocrystalline manganese borohydride (Mn (BH4) 2) after mechano-chemical synthesis and thermal dehydrogenation[J]. Acta materialia, 2015, 100: 392-400; Patri A, Umbreit T, Zheng J, et al. Energy dispersive X‐ray analysis of titanium dioxide nanoparticle distribution after intravenous and subcutaneous injection in mice[J]. Journal of Applied Toxicology: An International Journal, 2009, 29(8): 662-672; Lu P, Zhou L, Kramer M J, et al. Atomic-scale chemical imaging and quantification of metallic alloy structures by energy-dispersive X-ray spectroscopy[J]. Scientific reports, 2014, 4(1): 1-5.

  1. The quality of Figure 2 is quite poor. Would it be worth putting a better one? Or shrink it to make it look better?

Author reply: We thank the reviewer for raising this important concern. In this revised manuscript, we had replaced the original poor image in Fig. 2 with better one (see Fig. 1Sa in the revised manuscript).

                                         Fig. 1S

  1. Please make sure to check grammar and spelling in the next revision, "For exmaple" on page 1, for example.?

Author reply: We thank the reviewer for your valuabe comments. In the revised manuscript, we had checked the English expression carefully, including the wrong expression of “exmaple” in page 1.
